# A Potent PDK4 Inhibitor for Treatment of Heart Failure with Reduced Ejection Fraction

**DOI:** 10.3390/cells13010087

**Published:** 2023-12-30

**Authors:** Kenichi Aizawa, Akari Ikeda, Shota Tomida, Koki Hino, Yuuki Sugita, Tomoyasu Hirose, Toshiaki Sunazuka, Hiroshi Kido, Shigeyuki Yokoyama, Ryozo Nagai

**Affiliations:** 1Division of Clinical Pharmacology, Department of Pharmacology, Jichi Medical University, Shimotsuke 329-0498, Japan; stomida@jichi.ac.jp (S.T.); kh25322532@gmail.com (K.H.); 2Clinical Pharmacology Center, Jichi Medical University Hospital, Shimotsuke 329-0498, Japan; 3Division of Translational Research, Clinical Research Center, Jichi Medical University Hospital, Shimotsuke 329-0498, Japan; 4Ōmura Satoshi Memorial Institute, Kitasato University, Tokyo 108-8641, Japan; aikeda@lisci.kitasato-u.ac.jp (A.I.); sugitay2@sc.sumitomo-chem.co.jp (Y.S.); thirose@lisci.kitasato-u.ac.jp (T.H.); sunazuka@lisci.kitasato-u.ac.jp (T.S.); 5Division of Enzyme Chemistry, Institute for Enzyme Research, Tokushima University, Tokushima 770-8503, Japan; kido@tokushima-u.ac.jp; 6RIKEN Cluster for Science, Technology and Innovation Hub, Yokohama 230-0045, Japan; yokoyama@a.riken.jp; 7Jichi Medical University, Shimotsuke 329-0498, Japan

**Keywords:** heart failure, PDK4 inhibitor, TCA cycle, vitamin K_3_, 1,4-naphthoquinone, quinol monooxygenase

## Abstract

Heart failure with reduced ejection fraction (HFrEF) is characterized not only by reduced left ventricular ejection fraction (EF) but is also combined with symptoms such as dyspnea, fatigue, and edema. Several pharmacological interventions have been established. However, a treatment targeting a novel pathophysiological mechanism is still needed. Evidence indicating that inhibition of pyruvate dehydrogenase kinase 4 (PDK4) may be cardioprotective has been accumulating. Thus, we focused on vitamin K_3_ and used its framework as a new PDK4 inhibitor skeleton to synthesize new PDK4 inhibitors that show higher activity than the existing PDK4 inhibitor, dichloroacetic acid, and tested their cardioprotective effects on a mouse heart failure model. Among these inhibitors, PDK4 inhibitor **8** improved EF the most, even though it did not reverse cardiac fibrosis or wall thickness. This novel, potent PDK4 inhibitor may improve EF of failing hearts by regulating bioenergetics via activation of the tricarboxylic acid cycle.

## 1. Introduction

Heart failure with reduced ejection fraction (HFrEF) is characterized by a left ventricular ejection fraction under 40% [1,2], combined with symptoms such as dyspnea, fatigue, and edema [3]. Pharmacological interventions for HFrEF, including angiotensin-converting enzyme (ACE) inhibitors, angiotensin-receptor blockers (ARB), angiotensin receptor neprilysin inhibitor (ARNI), and beta blockers, improve ventricular function, reduce symptoms, and mortality and morbidity [4]. Yet, even with those pharmacological interventions, treatment that protects from acute heart failure, including acute exacerbation of HFrEF, is needed.

Pyruvate dehydrogenase (PDH) activates thiamine diphosphate (ThDP) [5]. Activated ThDP is capable of nucleophilic substitution on pyruvate, forming a ThDP–acetyl adduct [5]. The acetyl group is transferred to coenzyme A (CoA), forming acetyl CoA [5]. As a result, the tricarboxylic acid (TCA) cycle is activated, leading to ATP synthesis [5]. Pyruvate dehydrogenase kinase 4 (PDK4), one of the PDK isozymes, inactivates PDH by phosphorylating Ser293 and Ser300 of PDH, leading to the inactivation of the TCA cycle [6,7]. In failing hearts, the electron transport chain becomes impaired [8], resulting in greater reliance on glycolysis for energy production due to the attenuation of the TCA cycle [9]. PDK4 is highly expressed in hearts [6,10] and is upregulated in failing hearts [11], suggesting that PDK4 inhibition is a potential heart failure treatment.

Previous studies have shown that the amine salt of dichloroacetic acid (DCA), diisopropylamine dichloroacetic acid (DADA), which is used as a PDK4 inhibitor, can prevent metabolic disorders and multiorgan failure in severe influenza [12]. In a rat heart failure model, Dichloroacetic acid (DCA) has also been reported to improve cardiac function [13]. Nonetheless, the high IC_50_ of PDK4 inhibition (57.8 µM) and DCA’s neurotoxicity make it unsuitable for clinical application [14]. To establish a new, safe treatment for heart failure by inhibiting PDK4 activity, we searched for more active PDK4 inhibitors that are not structurally related to DCA or any other existing drugs. As a result, we found that Vitamin K_3_ has high PDK4 inhibition activity and created five novel PDK4 inhibitors derived from Vitamin K_3_ exhibiting potent PDK4 inhibition and low toxicity. We tested cardioprotection afforded by those PDK4 inhibitors.

## 2. Materials and Methods

### 2.1. Synthesis of Novel PDK4 Inhibitors

Detailed procedures for the synthesis of PDK4 inhibitors are provided in the Section 3 and Appendix A.

### 2.2. Assays for PDK4 Inhibition Capacity and Toxicity

#### 2.2.1. Description of PDK4

Full-length human PDK4 [amino acids 1-411 of accession number NP_002603.1] was expressed as an N-terminal Glutathione S-transferase (GST)-fusion protein (73 kDa) using an *E. coli* expression system. GST-PDK4 was purified using glutathione affinity chromatography.

#### 2.2.2. Preparation of Solution of Derivatives

Derivatives were dissolved in dimethyl sulfoxide (DMSO) and further diluted in DMSO to prepare a solution with a 100-fold higher concentration than the test sample solution. This solution was further diluted 25-fold in assay buffer (20 mM 4-(2-hydroxyethyl)-1-piperazineethanesulfonic acid (HEPES), 0.01% Triton X-100, 2 mM dithiothreitol (DTT), pH 7.5) to make a test sample solution. The positive control substance solution was prepared the same way.

#### 2.2.3. Off-Chip Mobility Shift Assay (MSA) (Carna Biosciences, Inc., Kobe, Japan) (Table 1)

(1) 5µL of test sample solution, 5 µL of substrate/ATP/metal solution in assay buffer (20 mM HEPES, 0.01% Triton X-100, 2 mM DTT, pH 7.5), and 10 µL of kinase solution in assay buffer prepared in assay buffer were mixed in the wells of a 384-well polypropylene plate and allowed to react for 5 h at room temperature. (2) The reaction was stopped by adding 70 µL of Termination Buffer (QuickScout Screening Assist MSA; Carna Biosciences). (3) Substrate peptides and phosphorylated peptides in the reaction solution were separated and quantified using the LabChip system (Perkin Elmer, Shelton, CT, USA). (4) The product ratio (P/(P+S)) was calculated from the substrate peptide peak height (S) and the phosphorylated peptide peak height (P) of the kinase reaction.

#### 2.2.4. Calculation

The average signal in control wells containing all reaction components was considered to be 0% inhibition. The average signal in background wells (without enzyme) was considered as 100% inhibition, and the % inhibition was calculated from the average signal in each test sample well.

#### 2.2.5. Cytotoxicity Assay against Human Fibroblast Cells

Wild-type fibroblasts obtained from a Japanese volunteer expressing [F352F + V368V] (FV–FV alleles), purchased from RIKEN Cell Bank (Tsukuba, Japan), were used, and the cytotoxicity assay was carried out as described in the previous paper [15].

### 2.3. Animals

C57BL/6J mice were purchased from Japan SLC, Inc. (Hamamatsu, Japan) and Takasugi Experimental Animals Supply Co., Ltd. (Kasukabe, Japan), and were kept under a 12 h light–dark schedule. All animal handling procedures in this study complied with the Jichi Medical University Guide for Laboratory Animals and the ARRIVE guidelines [16]. The Institutional Animal Care and Concern Committee at Jichi Medical University approved all experimental protocols. This study used 8-week-old male mice.

### 2.4. Transverse Aortic Constriction (TAC)

Mice were intubated prior to surgery and were maintained under assisted ventilation on a heating pad throughout the procedure. Under a microscope, a midline incision was made on the skin from the neckline to the mid-chest level. A partial thoracotomy was made through the second rib, and the sternum was retracted to aid visualization of the aorta. After periaortic tissue was removed, a 27-G blunt needle was placed in parallel along the transverse aorta, and the suture was tied over the needle between the innominate artery and the left common carotid. Then, the rib cage was closed, followed by closure of the skin. For the sham operation, the suture was tied in a manner that did not constrict the aorta. Mice were kept in a heating unit until they recovered completely from anesthesia.

### 2.5. Echocardiography

At weeks 0, 4, and 5 after TAC induction, mice were anesthetized with 1.5% isoflurane and secured on an animal handling and physiological monitoring system in the supine position. Images were captured with a 30 MHz probe attached to a Vevo 2100 (VisualSonics, Toronto, ON, Canada). Heart rate was maintained at 400 to 500 bpm, and the temperature of the system was kept at 37 °C while echocardiographic images were captured. The ejection fraction was calculated using the Vevo2100 system based on the Teichholz method.

### 2.6. Injection

Four weeks after the TAC surgery, mice with an EF less than 55 were subjected to injection. Then, 2.8 mg/kg of compound **8** or DCA in 5% DMSO in saline or 5% DMSO in saline was injected peritoneally every day for 7 days.

### 2.7. Sample Collection

Mice were euthanized 5 weeks after TAC induction (a week after the first injection) by injecting 100 mg/kg pentobarbital. Hearts were perfused with PBS containing PhosStop and cOmplete. Excised hearts were rinsed in phosphate-buffered saline (PBS) containing PhosStop and cOmplete. Approximately 50 mg of heart tissue was fixed in Ufix fixative and incubated overnight at 4 °C. Fixed heart specimens were stored in PBS at 4 °C until histological analysis. Approximately 20 mg of the heart was snap-frozen in liquid nitrogen and stored at −80 °C until the PDH activity assay.

### 2.8. Picrosirius Red Staining

Heart sections were de-paraffinized and rehydrated before being stained with 0.1% Sirius red in picric acid for 1 h. Then, sections were washed twice in 0.5% acetic acid. Sections were rehydrated and cleared before being sealed with Vectamount (Vector Laboratories, Newark, CA, USA). A BZ-9000 (Keyence, Osaka, Japan) was used to capture micrographs of heart sections. ImageJ [17] was used to measure fibrotic areas.

### 2.9. Immunoblot Analysis

Heart tissue samples were homogenized with a Bioprep-24 (Hangzhou Allsheng Instruments Co., Ltd.) in T-PER buffer (#78510, Thermo Fisher Scientific, Waltham, MA, USA), protease inhibitor cocktail (Roche Diagnostics, Basel, Switzerland), and phosphatase inhibitor cocktail (Roche Diagnostics). To collect supernatant, tissue lysates were centrifuged at 10,000 rpm for 10 min at 4 °C. Subsequently, 5 μg of protein were loaded in each lane of a 10% Bis–Tris gel (Thermo Fisher Scientific) and separated using SDS-PAGE and electroblotted to nitrocellulose membranes using an iBlot2 Dry Blotting System (Thermo Fisher Scientific). Membranes were blocked with Setsuyakukun supporter (DRC, Tama, Tokyo, Japan) for 1 h at room temperature, washed, and incubated for 2 h at room temperature in Kiwami Setsuyakukun buffer with anti-phospho-PDH antibodies (ABS204 and ABS194, Millipore), anti-total-PDH antibodies (ab110334, Abcam, Cambridge, United Kingdom), anti-PDK4 antibodies (ab89295, Abcam), or anti-GAPDH antibodies (AM3400, Thermo Fisher Scientific). Target proteins were detected with appropriate horseradish peroxidase-conjugated secondary antibodies (Cell Signaling Technology) and a chemiluminescence kit (#170-5060, Bio-Rad Laboratories). Signals were detected with a ChemiDoc Touch (Bio-Rad, Hercules, CA, USA), and quantitative analysis was performed using Image Lab (Bio-Rad).

### 2.10. PDH Activity

PDH Enzyme Activity Microplate Assay Kits (Abcam) were used to measure PDH activity. Detergent was added to 1/20 volume of heart lysates. The mixture was incubated on ice for 10 min and then centrifuged at 4 °C for 10 min at 1000× *g*. After centrifugation, the supernatant was collected. Samples were loaded into wells, gently mixed, and left at room temperature for 3 h. Then, samples in the wells were discarded, and 300 μL of 1× Stabilizer was loaded into each well and discarded. Before 200 μL of PDH assay solution was loaded into each well, 1× Stabilizer was loaded into the wells and discarded again. Absorbance was measured at 450 nm on a SpectraMax 340 (Molecular Devices, San Jose, CA, USA) for 30 min with a 20-second interval. After measuring absorbance, two points with the highest linearity were selected, and PDH activity was calculated using SoftMax Pro (Molecular Devices).

### 2.11. Statistical Analysis

The Mann–Whitney U test was used to compare distributed data between two groups. Comparisons of multiple groups that passed a Kolmogorov–Smirnov test for normality were performed with a one-way analysis of variance (ANOVA) using the Tukey post-hoc test.

## 3. Results

### 3.1. Synthesis of Derivatives of Vitamin K_3_ for PDK4 Inhibitors

In the search for a potent PDK4 inhibitor, we focused on *E. coli* quinol monooxygenase (QuMo) because of its high homology with PDK4. Comparisons of co-crystal structures of QuMo-Vitamin K_3_ (Figure 1a) and PDK4-ATP from the Protein Data Bank revealed that the ATP-binding pocket of PDK4 is very similar to that of QuMo [18]. The Vitamin K_3_ binding site in QuMo has a Glu residue, which interacts with the C1 carbonyl group of Vitamin K_3_, but the rest of the site is mostly composed of hydrophobic amino acid residues (Figure 1b). On the other hand, an Asp residue is located at the same position of the ATP-binding site in PDK4, but the rest of the site is composed of hydrophobic residues as in QuMo. This high homology suggested that Vitamin K_3_, whose crystal structure had already been reported with QuMo, would be a candidate inhibitor of PDK4. Based on this possibility, the PDK4 inhibition rate of Vitamin K_3_ was determined to be 66% at 10 µg/mL (Table 2), suggesting Vitamin K_3_ as a lead compound for novel PDK4 inhibitors. Therefore, we set out to create derivatives with higher PDK4 inhibitory activity based on the naphthoquinone structure, which is the basic skeleton of Vitamin K_3_.

To synthesize derivatives with highly potent PDK4 inhibition, we analyzed the binding model of 1,4-naphthoquinone, the core skeleton of Vitamin K_3_, to the ATP-binding site of PDK4 [18]. The skeleton fitted the site very well. The main interaction between the skeleton and the binding site was assumed to be hydrophobic binding because the neighboring amino acids are hydrophobic. Based on these results, we planned to introduce functional groups with various properties into the C6 and C7 positions of Vitamin K_3_ to produce highly inhibitory activity for PDK4. We established synthetic methods for the preparation of derivatives of Vitamin K_3_ that enhance its PDK4 inhibitory activity and water solubility, particularly C6- and C7-substituted derivatives. Especially, the C7-substituted 2-Me naphthoquinone derivative requires a multi-step reaction because the naphthoquinone ring has to be constructed using the benzoquinone derivative. Therefore, we selected a method to synthesize derivatives that can also be functionalized at the C5 position by changing the oxidation state of synthetic intermediates.

First, we focused on the derivatization of the C7 position of Vitamin K_3_ (Figure 1). According to the method of Maloney et al. [19], common intermediate **1** was synthesized in 5 steps from *o*-cresol. As shown in Figure 1, the Stobbe reaction of **1** with dimethyl succinate, followed by reduction with Pd/C, yielded **2** in 73% yield for two steps [20]. An intramolecular Friedel–Crafts reaction with polyphosphoric acid (PPA) led to cyclic **3** in 86% yield. Then, silane reduction of **3** quantitatively yielded decarboxylated product **4**. Finally, aromatic cyclization with DDQ and oxidation with CAN afforded **5,** possessing a methylcarbonyl group at the C7 position. In addition, we attempted to introduce a hydroxyl group at the C5 position to verify the effect of introducing a proton donor into **5**. The carbonyl group of **3** was reduced through the treatment of LDA with I_2_ to afford phenol **6**. The resulting hydroxyl group of **6** was then protected with an acetyl group and oxidized with CAN to form the 1,4-naphthoquinone skeleton **7**. Finally, the acetyl group was removed using HCl/MeOH to construct the C5,7-di-substituted derivative **8**.

On the other hand, derivatization of the C6 position of Vitamin K_3_ was achieved with a Friedel–Crafts reaction using the electron orientation of the C2 methyl group. According to known methods, including the Friedel–Crafts reaction, the acetylated product **9** was synthesized from Vitamin K_3_ in three steps [21]. Then, air oxidation of the methyl-ketone moiety in **9** under basic conditions produced carboxylic acid **10** in 72% yield [21,22] (Figure 2). Methyl esterification of **10** with HCl in MeOH, followed by oxidation with CAN, efficiently yielded the C6-methyl ester **11**. To further investigate the effect of the C6 position on inhibitory activity, **10** was subjected to condensation reactions with Gly-O*^t^*Bu and Ser(*^t^*Bu)-O*^t^*Bu. These condensations were advanced using PyBOP as a condensation reagent, affording **12a** and **12b**, respectively. Subsequent oxidation with CAN and deprotection of *^t^*Bu by treatment with TFA achieved the synthesis of **13a** and **13b,** with an amino acid side chain at the C6 position of Vitamin K_3_.

### 3.2. PDK4 Inhibitory Activity, Toxicity, and Cardioprotection of PDK4 Inhibitor Candidates

With five synthesized Vitamin K_3_ derivatives, their in vitro PDK4 inhibitory activity and cytotoxicity were evaluated. In vitro activities were tested using fluorescently labeled PDH by measuring phosphorylated and unphosphorylated PDH of the derivatives using a CCD camera. For cytotoxicity, IC_50_ values were calculated by adding the derivatives to the human fibroblast culture medium and monitoring growth inhibition. Then, we evaluated the efficacy of novel PDK4 inhibitor candidates in the treatment of heart failure. The transverse aorta of wild-type mice was constricted to induce a pressure overload in the left ventricle, and the ejection fraction (EF) was postoperatively measured at 4 weeks. Subsequently, 2.8 mg/kg of each derivative was intraperitoneally administered to mice once daily for 1 week to evaluate improvement in EF values.

As shown in Table 2, the PDK4 inhibitory activity of **5**, which has a methyl ester at the C7 position of Vitamin K_3_, was significantly attenuated compared to Vitamin K_3_. On the other hand, **8** with a hydroxyl group at the C5 position of **11** greatly enhanced PDK4 inhibitory activity but also enhanced cytotoxicity. PDK4 inhibitory activities of **11**, **13a**, and **13b**, in which a substituent was introduced at the C6 position of Vitamin K_3_, were similar to that of Vitamin K_3_. However, improvement of cytotoxicity was also observed, especially for **13a** and **13b**. Compound **8** showed the highest PDK4 inhibitory activity, which is 81.0% at 10 µg/mL. Calculated from the molecular weight of **8**, IC_80_ of **8** is 46 µM. In the previous study, the IC_50_ of dichloroacetic acid (DCA) for PDK4 inhibition was 57.8 µM [12]. Therefore, IC_50_ of **8** is assumed to be less than that of DCA. The effect of each derivative on EF values was examined, and all derivatives improved EF values compared to DCA (Figure 2). In particular, **8** showed the highest improvement, and we investigated further using **8**.

**Table 2 cells-13-00087-t002:** Inhibition capacity and toxicity of novel PDK4 inhibitors. The percentage inhibition of PDK4 at 10 µg/mL of each inhibitor is shown. IC_50_ values indicate half-maximal inhibitory concentrations required to inhibit fibroblast growth.

Compound	PDK4 Inhibitory Activity (%)	Toxicity (IC_50_ (µg/mL))
Vitamin K_3_	65.9	18
**5**	34.6	8
**8**	81.0	1
**11**	65.4	7
**13a**	66.8	>110
**13b**	63.2	>99

### 3.3. ***8*** Improves the Ejection Fraction

Next, we determined the dosage of **8**. We chose 2.8 mg/kg as the starting dose according to the dosage used in patent information for similar PDK4 inhibitors [23]. Administering half the dose of **8** (1.4 mg/kg/day) for 1 week starting 4 weeks after TAC did not increase EF compared to vehicle administration. The EF of mice given twice the dose, 5.6 mg/kg, had significantly higher EF than mice that received vehicle only but was approximately the same as that of mice that received 2.8 mg/kg of **8** (Appendix A). Thus, we determined that 2.8 mg/kg is the smallest effective dose. To assess its efficacy in treating heart failure, 2.8 mg/kg of **8** or vehicle was injected once a day into the peritoneal cavities of mice that had developed heart failure with reduced ejection fraction (EF < 55%). After 7 days, mice treated with **8** showed 1.5-fold higher EF than the vehicle-treated group (Figure 3a). After injection of **8**, 9 of 11 mice that received transverse aortic constriction (TAC) surgery showed increased EF. In contrast, the EF of vehicle-treated TAC mice either remained the same or decreased (Figure 3b). Echocardiography showed that only left ventricular internal dimensions at systole were significantly shorter in **8**-injected TAC mice (Figure 3c,d). Left ventricular internal dimensions at diastole, interventricular septum thickness at end-diastole and end-systole (IVS;d and IVS;s), and left ventricle posterior wall thickness at end-diastole and at end-systole (LVPW;d and LVPW;s) did not differ significantly between vehicle- and **8**-injected TAC mice (Figure 3c,d; Appendix A), suggesting that **8** improved cardiac function but did not reverse pathological cardiac remodeling. Furthermore, there was no significant difference in cardiac fibrosis between vehicle-injected TAC mice and KI079-injected TAC mice. Together, these results suggest that **8** improves contractility of failing hearts without reversing pathological cardiac remodeling.

### 3.4. Compound ***8*** Treats Heart Failure by Activating Pyruvate Dehydrogenase

We tested whether 7 days of daily treatment with **8** increased PDH activity of failing hearts. As expected, PDH activity was significantly increased after 7 days of treatments with **8** (Figure 4a). To confirm that the increase in PDH activity was due to the reduction of PDH phosphorylation via the inhibition of PDK4 activity, we tested the phosphorylation levels of PDH molecules. Western blotting analysis showed that phospho-PDH/total PDH ratios were lower in hearts from **8**-treated TAC mice than in hearts from vehicle-treated TAC mice (Figure 4b,c,e; Appendix A). Furthermore, protein expression of PDK4 was comparable in vehicle-treated and **8**-treated groups (Figure 4d,e; Appendix A), suggesting that **8** activates PDH by inhibiting PDK4 activity.

## 4. Discussion

At the start of this study, we focused on PDK4, which had long been expected to be a therapeutic target for severe influenza, and we had been researching PDK4 inhibitors that are not dependent on existing drugs [12]. Vitamin K_3_ has been reported to bind to *E. coli* quinol monooxygenase (QuMo), which is highly similar to the ATP-binding site of PDK4 [18]. Based on those reports, we hypothesized that Vitamin K_3_ is a PDK4 inhibitor and evaluated its PDK4 inhibitory activity. As a result, Vitamin K_3_ exhibited moderate inhibitory activity (66%) for PDK4 at 10 µg/mL.

Based on this result, modifications of the C6 and C7 positions of Vitamin K_3_ were performed to synthesize more active PDK4 inhibitors. Derivatization of the C7 position of Vitamin K_3_ utilized known methods to synthesize **5**, which has a methyl ester at the C7 position, and **8**, which also has a hydroxyl group at the C5 position of **5**. On the other hand, derivatization of the C6 position of Vitamin K_3_ was performed using the Friedel–Crafts reaction, which takes advantage of the orientation of the methyl group of Vitamin K_3_, to synthesize **11**, which has a methyl ester at the C6 position, and **13a** and **13b**, which were condensed with Gly and Ser after hydrolysis of the ester.

The heart expresses PDK4 highly, and during heart failure, PDK4 is upregulated [6,10,11]. Since exacerbation of both influenza severity and heart failure appear to converge on PDK4, we hypothesized that they have a common pathogenetic mechanism. Therefore, we conducted this study with the hypothesis that **8** treats heart failure.

In vitro PDK4 inhibitory activity, cytotoxicity (IC_50_), and improvement rates of left ventricular EF of the five synthesized derivatives were evaluated to investigate which compounds have the highest PDK4 inhibitory activity. Among these Vitamin K_3_ derivatives, **8** showed the highest PDK4 inhibitory activity in vitro but also had the highest cytotoxicity. On the other hand, **13a** and **13b** showed moderate PDK4 inhibitory activity, but their cytotoxicity was suppressed. Screening for EF improvement showed that **8** increased the EF of mice that received TAC, agreeing with the results of the experiment on PDK4 inhibitory capacity. Since **8**, which had the highest in vivo PDK4 inhibitory capacity, induced the highest EF recovery, the strength of PDK4 activity might predict the prognosis of heart failure.

We induced heart failure in mice using TAC and showed that daily injection of **8** for 7 days increased EF to the baseline level. Interestingly, only the left ventricular systolic diameter was decreased, but the thickness of the interventricular septae and the posterior walls did not change. Since fibrosis is often coupled with heart failure [24,25], we measured fibrotic areas. Heart sections from **8**-injected mice showed similar fibrotic levels to those of vehicle-injected mice. Thus, **8** did not reverse fibrosis, and the thickness of interventricular septae and posterior walls were similar in vehicle-injected TAC mice and **8**-injected TAC mice, suggesting that **8** improves contractility of failing hearts without significantly affecting cardiac physical properties. Energy insufficiency and metabolic alteration have been observed in failing hearts and have been proposed as a treatment target for heart failure [26,27]. Mechanisms of current treatment regimens can also be explained using energy availability [26]. ACE inhibitors reduce cardiac preload by decreasing peripheral resistance, and the renin–angiotensin–aldosterone system reduces energy consumption by hearts [26]. Beta-blockers spare energy by slowing the heart rate [26]. Since PDK4 was less active and PDH activity was preserved in **8**-injected TAC mice, we propose that improved bioenergetics via re-activation of the TCA cycle restore the pumping capacity of failing hearts. This supports the concept that energy insufficiency triggers heart failure. Since the cardioprotective effect of **8** was independent of cardiac remodeling, it is possible to treat heart failure regardless of preceding cardiovascular disease. Cardiac remodeling in heart failure with preserved ejection fraction (HFpEF) varies from patient to patient, and it is currently recommended that treatment be tailored to the patient’s condition [28]; however, PDK4 Inhibitor **8** could be administered regardless of individual cardiac remodeling. Furthermore, the smallest effective dose was determined to be 2.8/kg. This is equivalent to 0.228 mg/kg in humans. We did not compare the IC_50_ of fibroblast growth inhibition of **8** to that of DCA, although 2.8 mg/kg is less than 1/10 the clinical dose of DCA (25 to 100 mg/kg) [12]. Thus, even if the IC_50_ of fibroblast growth inhibition of DCA was the same as that of **8**, **8**’s low therapeutic dose makes **8** safer than DCA.

Hearts rely on the oxidization of fatty acids, glucose, and branched-chain amino acids (BCAAs) to produce enough energy [29], but BCAA accumulation is observed in hypertrophic and failing hearts [29,30]. Branched-chain α-keto acid dehydrogenase (BCKDH) produces substrates for the TCA cycle from BCAAs [29]. In a previous study, increasing BCAA catabolism through the inhibition of branched-chain ketoacid dehydrogenase kinase a week before TAC surgery preserved EF [31]. Inhibition of branched-chain ketoacid dehydrogenase kinase reduces phosphorylation of BCKDH and increases substrate production [32]. Thus, activation of BCAA catabolism may prevent EF reduction by increasing energy production via additional substrates for the TCA cycle. Likewise, Since PDK4 inhibition activates PDH, which produces acetyl-CoA, **8** may improve EF by reducing substrate insufficiency.

DCA improves cardiac function in heart failure patients and animals [13,33]. In addition to activation of the TCA cycle by enhancing PDK inhibition, DCA increases intracellular NADPH concentration by activating the pentose phosphate pathway, which is expected to protect against oxidative stress in failing hearts [13]. Since DCA is a PDK4 inhibitor, the pentose phosphate pathway could also be one of the pathways activated by **8.** Comprehensive metabolomic analysis of hearts in future studies will reveal the likely involvement of NADPH accumulation, BCAA catabolism, or regulation of other metabolites in the treatment of HFrEF using **8**.

Furthermore, a previous study showed that ACE inhibitors, ARNI, ARB, or beta-blockers improve the 2-year survival of heart failure patients [34]. However, some of the patients passed away in 2 years despite being treated with those medications [34]. Since PDK4 inhibition is a new treatment target for heart failure that existing treatment regimens do not cover, patients who respond poorly to those medications may be saved using compounds tested in the present study. The correlation of acetyl-CoA levels with cardiac functional improvement after treatment may also allow the identification of biomarkers that predict the prognosis and efficacy of treatment using PDK4 inhibition.

## 5. Conclusions

Structural analysis of PDK4 led to the development of a compound that treats HFrEF. Daily injection of a novel, potent PDK4 inhibitor treated HFrEF by increasing EF and decreasing left ventricular internal dimensions, but no other physical cardiac parameter. Thus, PDK4 inhibitor **8** may be used to treat heart failure regardless of cardiac remodeling. Furthermore, since PDK4 inhibition is a new treatment strategy, **8** is expected to increase the survival rate of patients who respond poorly to current treatment regimens.

## Data Availability

Datasets generated and/or analyzed during the present study are available from the corresponding author on request.

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
