# Peer review of "A Potent PDK4 Inhibitor for Treatment of Heart Failure with Reduced Ejection Fraction"

_cells, 2023, doi:10.3390/cells13010087_

Round 1

Reviewer 1 Report

Comments and Suggestions for Authors

cells-2748420, A Potent PDK4 Inhibitor for Treatment of Heart Failure with Reduced Ejection

by Kenichi Aizawa

Comments:

Abstract: 

-       The reviewer suggests including specific percentage for the “high mortality rate” to contextualize this claim. This helps in understanding the severity of the issue and the significance of the research.

-       The reviewer suggests briefly mentioning the methodology used for synthesizing and testing PDK4 inhibitors. This would add depth to the research's scientific rigor.

-       Could the author spell out the abbreviation 'TCA.'

Introduction: 

-       In line 36, "HErEF" appears to be a typographical error. It should be "HFrEF."

-       When mentioning high mortality despite existing treatments (line 39), it would be impactful to include percentage to support this claim.

-       The reviewer suggests adding a brief description of the biochemical processes involving pyruvate dehydrogenase (PDH), the tricarboxylic acid (TCA) cycle, and PDK4 (lines 41-45).

-       The phrase "tricarboxylic acid cycle (TCA) cycle" is redundant. It should be "tricarboxylic acid (TCA) cycle" (lines 44 and 45).

-       linking DCA’s limitations (lines 47-49) more directly to the rationale for synthesizing novel PDK4 inhibitors could strengthen the narrative.

-       The reviewer suggests expanding slightly on how the novel PDK4 inhibitors were selected or designed based on the limitations of DCA to the rationale for your study (lines 50-52).

-       Ensure that all abbreviations are spelled out in full when they first appear in the text. For example, when you introduce terms like "ACE inhibitors" or "pyruvate to acetyl CoA," please provide the full forms of any abbreviations used in these contexts initially.

Material and methods: 

-       It's mentioned that the detailed procedures for the synthesis of PDK4 inhibitors are provided in the Supplementary information (line 56). It might be helpful to briefly summarize the key steps or novel aspects of the synthesis process in the main text for readers who might not refer to the supplementary materials.

-       When abbreviations like "DMSO" or "TAC" are first introduced (lines 63 and 85, respectively), it's important to spell them out in full when first mentioned. 

-       Ensure consistent use of terminology throughout the section. For instance, the term "PDK4" is sometimes referred to as "PDHK4" (lines 59 and 75). Consistency is key for clarity and professionalism.

-       It's important to state that animal experiments were conducted in accordance with relevant guidelines and regulations, and that necessary ethical approvals were obtained.

Results:

Synthesis of derivatives of Vitamin K3 for PDK4 inhibitors:

-       It might be helpful to provide a bit more context on how the PDK4 inhibition rate of Vitamin K3 compares to existing inhibitors or why this level of inhibition is promising.

-       Ensure consistent use of the abbreviation for PDK4. In line 180, it is mistakenly written as "PKD4."

PDK4 inhibitory activity, toxicity, and cardioprotection of PDK4 inhibitor candidates:

-       The reviewer suggests adding a bit more detail on the methodology, such as the specific conditions of the fibroblast culture.

-       Details on how growth inhibition was monitored are important. This could include the time points of measurement, the methods used to assess cell viability or proliferation (such as specific assays), and how these measurements were quantified.

-       An explanation of how the IC50 values were calculated would be beneficial. This should include the method used for data analysis, such as the software or mathematical models, and how multiple experiments were averaged or statistically analyzed.

8 improves the ejection fraction:

-       The in vivo testing of the derivatives, particularly compound 8, is an essential part of your study. Clarify why 2.8 mg/kg was chosen as the dosage and how this relates to potential human applications. Also, a bit more detail on the echocardiography measurements would be beneficial.

-       Please ensure to include the representative Western blot images of phospho-PDH and total PDH in Figure 4, as they are currently missing.

Discussion: 

-       The initial mention of the possibility that PDK4 could be useful in treating severe influenza seems a bit disconnected from the rest of the discussion, which focuses on heart failure. If there's a direct link between influenza and heart failure in the context of PDK4 inhibition, it should be clarified. If not, consider focusing solely on heart failure to maintain coherence.

-       Since the abbreviation 'EF' for ejection fraction has already been defined earlier in the manuscript, it is not necessary to repeatedly spell out its full description in subsequent sections. The same for ' TAC.'

-       The discussion effectively balances in vitro and in vivo results. However, expanding on how the in vitro findings (like cytotoxicity and PDK4 inhibition) translate into the observed in vivo effects would deepen the understanding.

-       It would be beneficial to discuss further how the mechanism by which compound 8 improves heart function fit into the broader context of current heart failure treatments and research.

-       The comparison with other pathways and treatments, such as DCA and BCAA metabolism, is insightful. This section could be strengthened by discussing how compound 8 might be integrated into existing treatment regimes or how it compares in terms of efficacy and safety.

-       Clarify what specific aspects or metabolites might be of interest in the future studies and how they could potentially influence the understanding or treatment of HFrEF.

-       The summary might be enhanced by briefly mentioning the potential clinical implications or the next steps in developing compound 8 as a therapeutic agent.

Comments on the Quality of English Language

Minor editing of English language required

Reviewer 2 Report

Comments and Suggestions for Authors

In the article "A Potent PDK4 Inhibitor for Treatment of Heart Failure with  Reduced Ejection Fraction" authors concentrate on study of new potential factor that can be used in future Heart Failure with  Reduced Ejection Fraction treatment. This novel agent acts by use of alternative metabolic pathway, not present in nowadays heart failure therapy. So the article is very original, highly valuable and can have great impact on future heart failure therapy development. I recommend strongly to accept the article. The Authors should consider very little changes:

1. First sentence from discussion:" In this study, we focused on the possibility that PDK4 could be a useful drug in the 317 treatment of severe influenza, and decided to proceed with exploratory research for PDK4 318 inhibitors without existing drugs" should be clearly explained

2.Possible clinical impact of potential new drug on different heart failure populations should be underlined in discussion. This is especially interesting in the light of fact of lack of influence of PKD4 inhibitors on structural changes in heart however potential improving the metabolic function.

3. More information if possible about potential data of cell toxicity  of PKD4 inhibitors can be mentioned.

4. Deeper comparision with different mentioned factors (DCA, BCAA) can be made.

Round 2

Reviewer 1 Report

Comments and Suggestions for Authors

The reviewer would like to thank the authors for addressing all the concerns raised in the first round of revision.

Comments on the Quality of English Language

Minor editing of English language required